

# Most ornamental plants on sale in garden centres are unattractive to flower-visiting insects

Mihail Garbuzov, Karin Alton and Francis L.W. Ratnieks

Laboratory of Apiculture & Social Insects, School of Life Sciences, University of Sussex, Falmer, Brighton, East Sussex, United Kingdom

## ABSTRACT

**Background**. Gardeners and park managers seeking to support biodiversity in urban areas often plant ornamentals attractive to flower-visiting insects. There is a huge diversity of garden plant varieties, and some recommendations are available as to which are attractive to insects. However, these are largely not based on rigorous empirical data. An important factor in consumer choice is the range of varieties available for purchase. In the UK, garden centres are a key link in the supply chain between growers and private gardens. This study is the first to determine the proportions of flowering ornamentals being sold that are attractive to flower-visiting insects.

**Methods**. We surveyed six garden centres in Sussex, UK, each over two days in 2015, by making 12 counts of insects visiting patches of each ornamental plant on display for sale that was in bloom. To provide a consistent baseline among different locations, we brought with us and surveyed marjoram (*Origanum vulgare*) plants in pots, which are known to be attractive to a wide range of flower-visiting insects. The attractiveness of plant varieties to insects was then expressed in two ways: the absolute number and relative to that on marjoram ('marjoram score'), both per unit area of plant cover. In addition, we noted whether each variety was recommended as pollinator-friendly either via a symbol on the label, or by being included in the Royal Horticultural Society's 'Perfect for Pollinators' list. Furthermore, we compared the attractiveness of plants that are typically grown for more than one year versus only one year.

**Results**. We surveyed 59–74 plant varieties in bloom across the six garden centres. In each garden centre, the distributions of variety attractiveness were highly skewed to the right, with most varieties being relatively unattractive, and few varieties highly attractive to flower-visiting insects. The median attractiveness of varieties with a recommendation was 4.2× higher than that of varieties without. But, due to the large variation there was a substantial number of both poor varieties that had a recommendation and good varieties that did not. Median attractiveness of multi-year plants was 1.6× that of single-year plants, with a similar overlap in distributions.

**Discussion**. Our study demonstrates the practicality of carrying out plant surveys in garden centres. Garden centres display large numbers of varieties for sale, most of which are in bloom. Furthermore, data gathered in garden centres appear to correlate well with data gathered in two previous studies in Sussex for plants established in gardens. Although it is unclear whether the varieties being sold in garden centres are a fair representation of varieties that are actually grown by gardeners, our results suggest that there might be considerable scope for making parks and gardens considerably more insect-friendly through judicious variety choices.

Corresponding author
Mihail Garbuzov,
m.garbuzov@sussex.ac.uk,
mihail.garbuzov@gmail.com

## INTRODUCTION

Pollinating insects appear (*Ghazoul, 2015*) to be in global decline, with one of the main causes being the loss and degradation of habitats resulting in loss of floral resources (*Potts et al., 2010*; *Kennedy et al., 2013*; *Goulson et al., 2015*). Urban areas, including ornamental plants in parks and gardens, could play an important role in providing nectar and pollen for flower-visiting insects (*Goddard, Dougill & Benton, 2010*; *Baldock et al., 2015*; *Salisbury et al., 2015*). Numerous ornamentals are recommended as ''friendly'' to bees, butterflies and other pollinators, with multiple prominent conservation organizations compiling lists or labelling plants being sold to the public as pollinator-friendly (*Garbuzov & Ratnieks, 2014a*). However, in most cases, these recommendations appear not to be based on empirical evidence (*Garbuzov & Ratnieks, 2014a*). A further difficulty in compiling such lists is the fact that many garden plants have many varieties which can vary greatly in their attractiveness to flower-visiting insects. For example, asters are often recommended as pollinator-friendly, while there are hundreds of varieties, which range from highly attractive to non-attractive (*Garbuzov & Ratnieks, 2015*).

The capacity of urban areas to provide food for flower-visiting insects is affected by the choices of gardeners and garden managers. For example, a survey in one public park in a small town in southern England showed that only a small proportion, 4%, of the 79 varieties in bloom were highly attractive to flower-visiting insects, most of which were bees and hover flies (*Garbuzov, Samuelson & Ratnieks, 2015*). No insects at all were seen on 30% in 15 counts. This suggests there is scope for making parks and gardens more bee- and insect-friendly. In addition, this can be done at zero cost as pollinator-friendly varieties are not more expensive, more difficult to grow or obtain, or less attractive to the human eye (*Garbuzov & Ratnieks, 2015*). Furthermore, *Lindemann-Matthies & Marty (2013)* have shown that even on the scale of a whole garden, species-rich gardens of high ecological quality tend to be perceived as more aesthetically pleasing by humans.

In the UK, 'wildlife gardening' is actively promoted by conservation organizations (*Ryall & Hatherell, 2003*), and most private gardeners engage in some form of it (*Mew et al., 2003*). Motivations for doing so are varied, but notably include a sense of personal wellbeing and moral responsibility for nature (*Goddard, Dougill & Benton, 2013*). Indeed, biodiverse nature in urban areas does have a positive influence on physical, psychological and social wellbeing (*Clark et al., 2014*; *Shanahan et al., 2015*).

Garden centres are a key part of the supply chain delivering plants from wholesale growers to private gardens. Indeed, garden centres have become a prominent part of the British shopping and leisure culture. For instance, the largest UK garden centre chain, Wyevale, which operates 153 branches nationwide, is estimated to have attracted 46 million visitors and reported an annual revenue of £311 million in 2015 (*Wyevale, 2015*). In addition to selling plants, garden centres typically offer a wide range of other goods.

These primarily include products associated with gardening, such as gardening tools and equipment, compost, seeds, clothing, as well as outdoor garden furniture, barbeques, and wildlife-gardening products, such as bird houses and "bee hotels". However, many garden centres have expanded into other household niches, such as food, gifts, clothes, shoes, and even into pet and aquarium products. Almost all garden centres now have restaurants or cafés, as customers tend to make long visits and enjoy lunch or coffee breaks in the interim. In short, garden centres have become a specialized type of shopping centre or mall aimed at attracting customers interested in products for home and garden, for a drink, snack or meal, or for a social outing.

Many gardeners obtain plants from garden centres and similar organizations. As a result, a key factor in the friendliness of gardens to flower-visiting insects is likely to be the plant varieties on sale to the general public. Here, we present the first study of this link in the supply chain by surveying ornamental plants in six retail locations in southern England and quantifying their attractiveness to flower-visiting insects. Garden centres are also practical locations for surveying ornamental plants, as most varieties are sold when they are at or near full bloom, presumably to make them more attractive to customers. Additionally, we determined whether the varieties on sale that had pollinator-friendly symbols on their labels or were included on the UK's Royal Horticultural Society 'Perfect for Pollinators' list (*RHS, 2014*) are more attractive to flower-visiting insects than non-promoted varieties; and whether hardy multi-year plants are more attractive than single-year bedding plants. Quantifying the attractiveness of plants available to the public, and understanding how this depends on promotional labelling will hopefully lead to introducing changes that could have tangible benefits for insect conservation in urban areas.

## MATERIALS & METHODS

### Garden centres

Six retail locations (five garden centres and one nursery, hereafter 'garden centres', Table 1) were surveyed between 9 July and 19 September 2015. All six garden centres were in East Sussex, southern England, UK, within c. 30 km of each other. The garden centres were chosen to reflect the diversity of retail types, and included three independent businesses, each with a single site, and three that were branches belonging to medium or large chains. The independents propagated and grew some or all of their stock, while the chains retailed stock brought in from wholesale suppliers.

### Survey methodology

In each garden centre we only surveyed those plant varieties that were at or near peak bloom and were displayed for sale to the customers. This was between 59 and 74 varieties covering between 14.7 and 37.6 m$^2$ per garden centre (Table 1). The plants were all being sold in pots and located in the outdoor areas, making them fully accessible to foraging insects. Plant varieties were arranged in groups forming discrete patches made up of a small number of pots, with an average area of 0.23–0.55 m$^2$ per garden centre. Previous research has found that flower patch size does not affect the number of insects attracted

Garbuzov et al. (2017), *PeerJ*, DOI 10.7717/peerj.3066

Peerj

**Table 1** **General information on the five garden centres and one nursery surveyed, and the plant information summary.**

| Name | Garden Pride | Wyevale | Staverton | Marchants | Hillier | Wych Cross |
|---|---|---|---|---|---|---|
| **(A) Business information** | | | | | | |
| Address | Common Lane, Ditchling, BN6 8TP | A22 Lower Dicker, Hailsham, BN27 4BJ | Eastbourne Road, Halland, BN8 6PU | 2 Marchants Cottages, Mill Lane, Laughton, BN8 6AJ | Hailsham Road, Stone Cross, Pevensey, BN24 5BS | Forest Row, RH18 5JW |
| Dates surveyed in 2015 | 08–09 Jul | 14–15 Jul | 29–30 Jul | 07–08 Aug | 09–10 Sep | 18–19 Sep |
| Type of business | Medium chain garden centre | Large chain garden centre | Independent garden centre | Independent nursery and garden | Medium chain garden centre | Independent[a] garden centre |
| Total number of stores | 18 | 153 | 1 | 1 | 12 | 1 |
| **(B) Plant information** | | | | | | |
| Varieties surveyed | 68 | 69 | 68 | 74 | 59 | 66 |
| Area of varieties surveyed ($m^2$) | 37.6 | 34.1 | 30.4 | 16.9 | 14.7 | 27.6 |
| Proportion with recommendation[b] | | | | | | |
|   RHS Perfect for Pollinators list | 46% | 38% | 59% | 62% | 59% | 59% |
|   Symbol on label | 18% | 10% | 7% | 0% | 29% | 24% |
|   No recommendation | 49% | 62% | 41% | 38% | 37% | 39% |
| Plant varieties by type[c] | | | | | | |
|   Multi-year | 88% | 46% | 88% | 99% | 83% | 92% |
|   Bedding | 12% | 54% | 12% | 1% | 17% | 8% |
| Overall garden centre attractiveness | | | | | | |
|   Mean marjoram score[d] per variety | 0.12 | 0.25 | 0.20 | 0.32 | 0.36 | 0.33 |
|   Median marjoram score[d] of varieties | 0.04 | 0.06 | 0.11 | 0.20 | 0.18 | 0.18 |
|   Better than marjoram | 1% | 7% | 0% | 7% | 7% | 5% |

**Notes.**

[a] Shortly after our survey, in October 2015, the independent Wych Cross garden centre was acquired by the Wyevale garden centre chain.

[b] Recommendations by the RHS list and symbols on pots or labels overlapped, so the three proportions do not add up to 100%.

[c] Multi-year plants included fully hardy perennials, biennials and shrubs; bedding plants included annuals, as well as half-hardy and tender biennials and perennials.

[d] Insect count per $m^2$ relative to marjoram, *Origanum vulgare*, two patches of which were brought in and set up by the researchers at each location as a benchmark.

per unit area in the range 0.1–3.1 m$^2$ (*Garbuzov, Madsen & Ratnieks, 2015*), which makes it possible to make unbiased comparisons of varieties in patches of different areas.

In each garden centre, the survey took two days. On day 1, we made a list of all suitable plant varieties present and measured their patch areas. On day 2, we made 12 counts of insects visiting patches of each variety between 10:00 and 16:30 BST at 30-minute intervals. At each count, the insects foraging in each patch were identified by eye, without capture, into the following categories: (i) honey bee, *Apis mellifera*, (ii) bumble bees, *Bombus* spp., and sub-groups after *Fussell & Corbet (1992)*, (iii) other bees, non-*Apis* and non-*Bombus* Anthophila, (iv) hover flies, Diptera: Syrphidae, (v) other flies, non-syrphid Diptera, (vi) butterflies & moths, Lepidoptera, with butterflies identified to species, (vii) wasps, both social and solitary species and (viii) all other insects. The attractiveness of plants to insects was then expressed as the mean number of insects per count per m$^2$. When plants being surveyed were purchased by customers during the course of the count day, the patch area was re-measured and the calculation adjusted accordingly.

## Marjoram baseline

To provide a consistent baseline across the six garden centres, we also surveyed two flowering patches of marjoram, *Origanum vulgare*, each with an area of 0.45 m$^2$ and consisting of six 10 L pots. We brought these plants in ourselves and set them up the day before counting began to allow time for insects to discover them. The two marjoram patches were included in the counting routine with all other plant varieties. In this way, the attractiveness of the other varieties could also be expressed relative to marjoram, hereafter the 'marjoram score'. Marjoram was chosen as it is highly attractive to a wide range of flower-visiting insects (*Garbuzov & Ratnieks, 2014b*; *Shackleton & Ratnieks, 2016*).

## Pollinator-friendly recommendations and plant longevity

We determined whether each variety (a) was present on the UK's Royal Horticultural Society's 'Perfect for Pollinators' list (*RHS, 2014*), one of the longest and most authoritative lists of its kind (*Garbuzov & Ratnieks, 2014a*), or (b) had any bee- or pollinator-friendly symbol or logo on the pot or label, including the RHS Perfect for Pollinators logo (Fig. 1). There was some overlap between the two categories.

In addition, we classified plant longevity into two categories: (a) multi-year: plants that survive for more than one season in Britain, which includes shrubs (e.g., *Buddleja*, *Lavandula*) and fully-hardy herbaceous perennials (e.g., *Helenium*, *Salvia*) and biennials (e.g., *Digitalis*); (b) bedding plants: these typically survive for only one season, and include annuals (e.g., *Petunia*, *Viola*) plus half-hardy and tender perennials (e.g., *Pelargonium*, *Dahlia*) and biennials (e.g., *Rudbeckia hirta*). Full hardiness was defined as hardiness through most of the UK, corresponding to the RHS hardiness rating category H4 (down to −10 to −5 °C) (*RHS, 2016*).

## RESULTS

### Relative abundance of insect groups

In total 325–2,339 insects were counted per garden centre (Table 2). The overall insect profile varied among garden centres, but was generally similar to what was previously

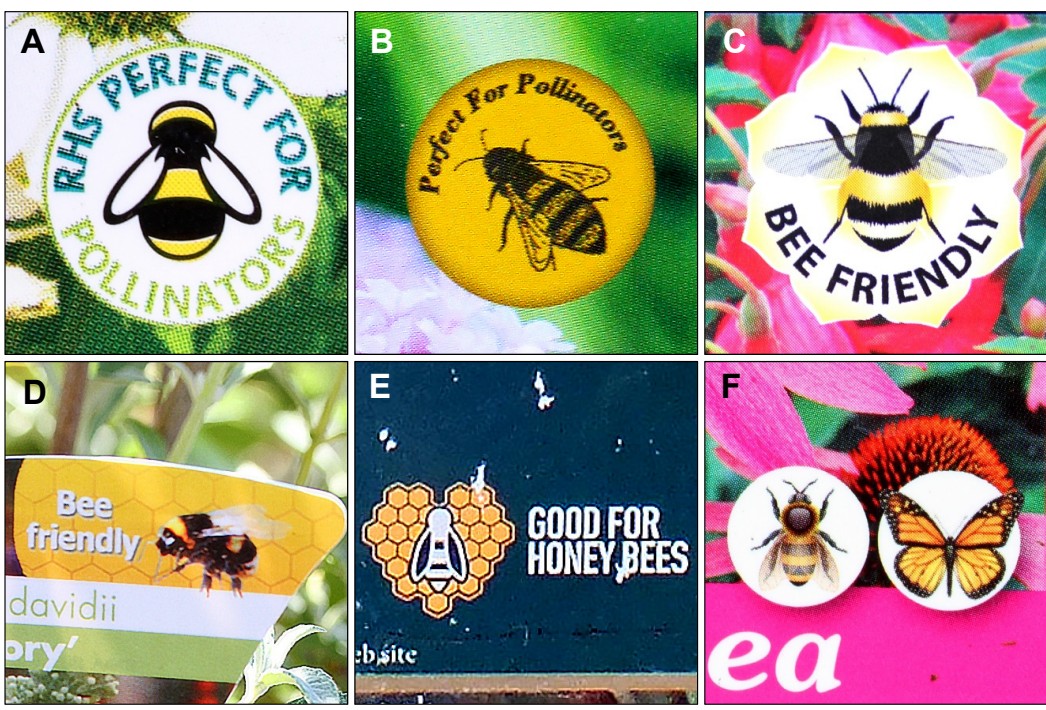

**Figure 1** **Examples of some pollinator-friendly logos found on plant labels in garden centres.** (A) RHS Perfect for Pollinators, (B) non-RHS Perfect for Pollinators, (C, D) Bee friendly, (E) Good for honey bees, (F) Bee and butterfly images.

recorded on plants growing in gardens in East Sussex (*Garbuzov & Ratnieks, 2014b*; *Garbuzov, Samuelson & Ratnieks, 2015*). Bees were the most abundant group (mean 50.9%, range 16.9–64.7%, Table 2) in five of the six garden centres, followed by flies (mean 43.1%, range 28.0–78.6%), of which syrphid flies were always more common than the non-syrphid flies. Butterflies and moths were generally less abundant (mean 3.2%, range 0.1–14.5%), as were wasps (mean 2.7%, range 0.2–11.3%). Other insect types were recorded in only one of six garden centres, comprised only 0.8% of the counts (Wyevale) and were all longhorn beetles (Coleoptera: Cerambycidae).

## Consistency with previous studies

The attractiveness of five plant species or genera that were common between this study and two previous studies of plants in gardens in East Sussex was in broad agreement (Table S1; Fig. 2). The percentile rank recorded in this study correlated significantly with that in a previous study of 32 plant varieties on the University of Sussex campus in Falmer ($r = 0.92$, $P = 0.026$, *Garbuzov & Ratnieks, 2014b*) and a survey of 79 plant varieties in a park in Lewes ($r = 1.00$, $P = 0.004$, *Garbuzov, Samuelson & Ratnieks, 2015*). Similarly, mean insect count per $m^2$ on the same plants in this study correlated significantly with that in the previous study in Falmer ($r = 0.99$, $P = 0.001$, *Garbuzov & Ratnieks, 2014b*) and marginally non-significantly in the previous study in Lewes ($r = 0.91$, $P = 0.089$, *Garbuzov, Samuelson & Ratnieks, 2015*).

**Table 2 Finer breakdown of flower-visiting insect counts in the eight main categories.**

| | Garden Pride | Wyevale | Staverton | Marchants | Hillier | Wych Cross |
|---|---|---|---|---|---|---|
| **Bees** | | | | | | |
| Honey bees, *Apis mellifera* | 49 | 2 | 36 | 812 | 508 | 576 |
| Bumble bees, *Bombus* spp.[a] | | | | | | |
|    *B. terrestris/lucorum* group | 88 | 53 | 524 | 13 | 40 | 42 |
|    *B. hortorum* group | 6 | – | 1 | 7 | – | 5 |
|    *B. pratorum* group | 4 | – | 5 | 8 | – | 0 |
|    *B. lapidarius* group | – | – | 10 | – | – | 0 |
|    *B. pascuorum* group | 23 | 3 | 43 | 19 | 86 | 416 |
|    All bumble bees | 121 | 56 | 583 | 47 | 126 | 463 |
| Other bees | 16 | 5 | 16 | 79 | 28 | 216 |
| All bees | 186 | 63 | 635 | 938 | 662 | 1,255 |
| **Flies** | | | | | | |
| Hover flies | 79 | 212 | 284 | 374 | 308 | 639 |
| Other flies | 12 | 81 | 34 | 238 | 146 | 396 |
| All flies | 91 | 293 | 318 | 612 | 454 | 1,035 |
| **Butterflies & moths** | | | | | | |
|    *Thymelicus sylvestris* | 4 | – | – | – | – | – |
|    *Ochlodes sylvanus* | 1 | – | – | – | – | – |
|    *Pieris rapae* | – | – | 2 | – | – | – |
|    *Pieris brassicae* | – | – | 1 | – | – | – |
|    *Aglais io* | – | – | 1 | – | – | – |
|    *Aglais urticae* | 40 | – | – | – | – | – |
|    *Vanessa atalanta* | – | – | – | – | 1 | 6 |
|    *Polygonia c-album* | 1 | 1 | – | 3 | – | – |
|    *Maniola jurtina* | 1 | 11 | – | – | – | 2 |
|    *Pararge aegeria* | – | – | – | – | – | 6 |
|    Moths | – | 1 | – | – | – | – |
|    All butterflies & moths | 47 | 13 | 4 | 3 | 1 | 14 |
| **Wasps** | | | | | | |
|    Social wasps, Hymenoptera: Vespinae[b] | – | – | 23 | 182 | 1 | 15 |
|    Other wasps, non-social apocrite Hymenoptera, excluding bees, ants and sawflies | 1 | 1 | 1 | 15 | 1 | 20 |
| **Other** | | | | | | |
|    Beetles, Coleoptera | – | 3 | – | – | – | – |
|    All insects | 511 | 436 | 1,616 | 2,688 | 1,781 | 3,594 |

**Notes.**
[a]Bumble bee identification groups follow *Fussell & Corbet (1992)*.
[b]Social wasp species are not confined to Vespinae, but vespine wasps are the only social wasps that occur in Britain.

## Most varieties attracted few or no insects, few varieties were very attractive

In all six garden centres most varieties on sale were relatively unattractive to insects. Frequency distributions of variety attractiveness in each garden centre were highly positively skewed, with most varieties (mean 68%, range 33–97%) attracting 0–5 insects per count

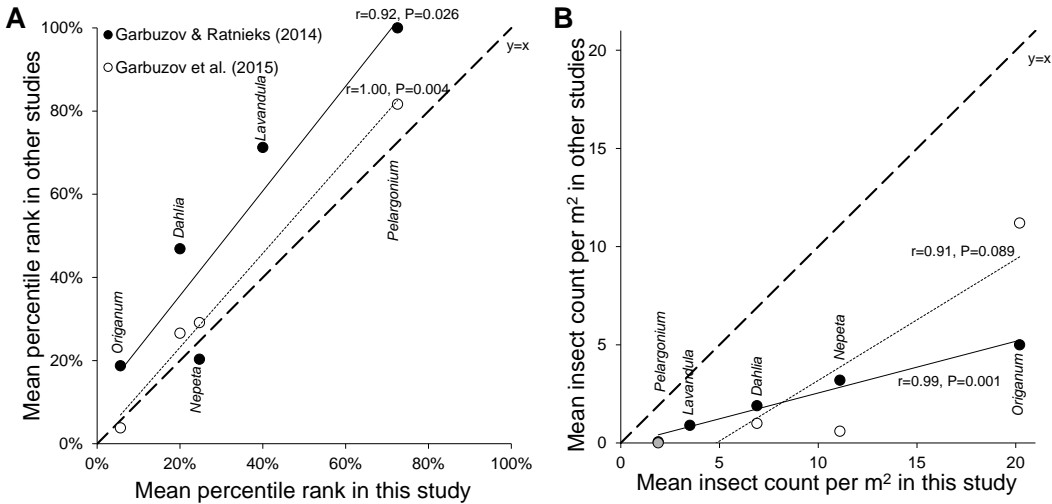

**Figure 2  Consistency with previous studies.** Correlations of (A) percentile rank and (B) mean insect count per m² between plant varieties surveyed in this study and two previous studies conducted in Falmer (closed circles and solid line, *Garbuzov & Ratnieks, 2014b*) and Lewes (open circles and dotted line, *Garbuzov, Samuelson & Ratnieks, 2015*), all in East Sussex, England. Dashed line is the identity line, $y = x$.

per m² (Fig. 3). Fewer varieties (mean 13%, range 1–22%) attracted 5–10 insects or >10 insects (mean 19%, range 1–48%) per count per m².

When attractiveness is expressed as marjoram scores the pattern is similar. On average 67% (range 53–88%) of varieties had marjoram scores between 0–0.25 (Fig. 3), 18% (range 7–33%) between 0.25–0.5 and 15% (range 4–24%) greater than 0.5. On average, only 4% (range 0–7%) of varieties had a marjoram score greater than 1, i.e., were more attractive than marjoram.

### Are recommended varieties better than non-recommended?

Marjoram scores were significantly associated with whether or not the plants have been recommended as bee- or pollinator-friendly (Kruskal–Wallis test, $K = 54.99$, $df = 2$, $P < 0.001$, Fig. 4). *Post-hoc* pairwise comparisons showed that both (1) varieties on the RHS Perfect for Pollinators list (median = 0.21, IQR: 0.05-0.46; Wilcoxon–Mann–Whitney test, $U = 11668.00$, $P < 0.001$) and (2) varieties with a bee- or insect-friendly symbol on pot or label (median = 0.21, IQR: 0.06–0.40, Wilcoxon–Mann–Whitney test, $U = 7377.50$, $P < 0.001$) were more attractive than varieties with no recommendation (median = 0.05, IQR: 0.00-0.15), corresponding to a median increase of 320%. However, varieties in the two recommended categories were not different from each other (aggregate median = 0.21, aggregate IQR: 0.05–0.45, Wilcoxon–Mann–Whitney test, $U = 6138.00$, $P = 0.931$).

### Are multi-year plants more attractive than bedding plants?

Multi-year plants (median = 0.13, IQR: 0.03–0.37) had significantly higher marjoram scores than bedding plants (median = 0.08, IQR: 0.00–0.17; Wilcoxon–Mann–Whitney test, $U = 13986$, $P = 0.006$, Fig. 5), corresponding to a median increase of 63%. However, long tails in the distributions of both plant groups meant that most multi-year plants (320 of 335, 96%) and bedding plants (66 of 69, 96%) were less attractive than the marjoram baseline.

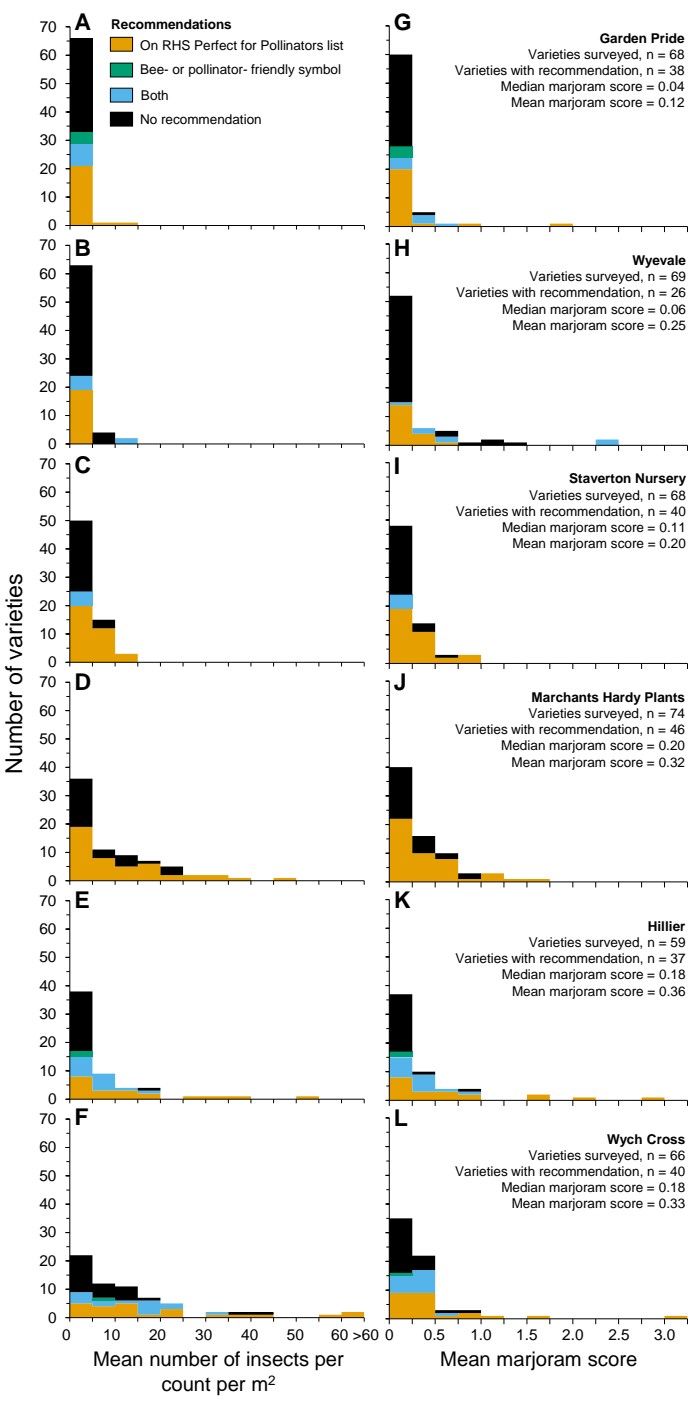

**Figure 3** Frequency distributions of insect-attractiveness of plant varieties in the six garden centres surveyed, based on the mean numbers of insects per count per $m^2$ (A–F) and on the mean counts per $m^2$ relative to marjoram, *Origanum vulgare* (G–L). Proportions of varieties in each bin are coloured according to recommendation: RHS Perfect for Pollinators list (orange), bee- or pollinator-friendly symbol on pot or label (bluish green), or both (sky blue); remaining varieties without a recommendation are coloured in black.

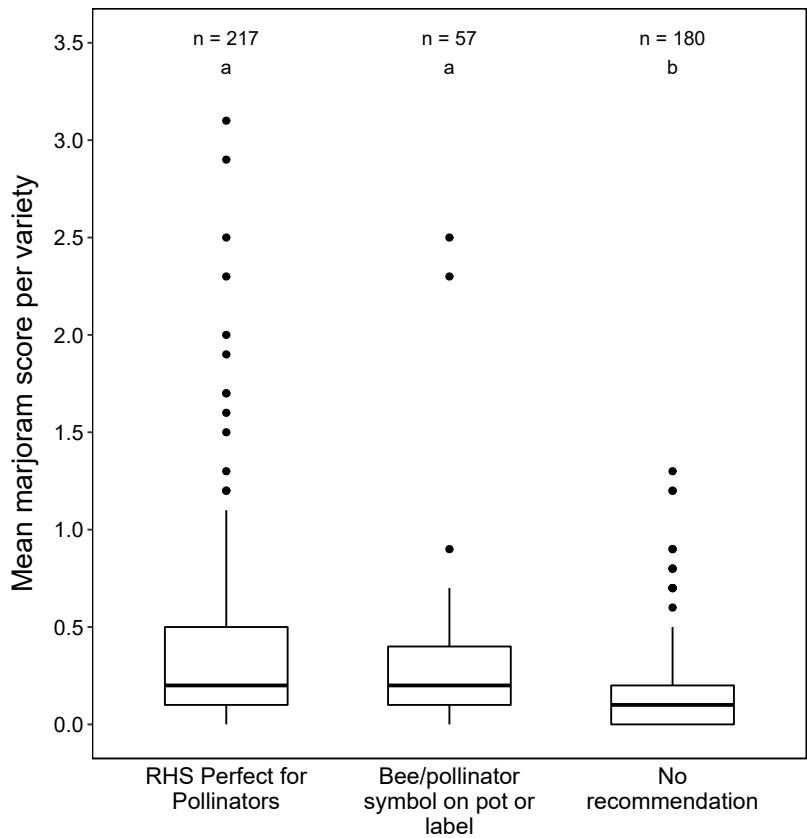

**Figure 4** **Boxplots of mean marjoram scores of varieties according to recommendation category.**
Lowercase letters above boxplots denote homogeneous subsets based on pairwise comparisons using the
Wilcoxon–Mann–Whitney median comparison test ($\alpha = 0.05$), where categories not sharing a letter are
significantly different.

## DISCUSSION

### Most varieties were relatively unattractive to flower-visiting insects

The majority of plant varieties in each garden centre were relatively unattractive to
flower-visiting insects, with the frequency distributions of attractiveness strongly skewed
to the right (Fig. 3). Similar patterns were seen in the distributions of both actual insect
counts and marjoram scores. These results are also very similar to those obtained in
an earlier survey of 79 plant varieties in a public park in the same region (*Garbuzov,
Samuelson & Ratnieks, 2015*) indicating that this pattern is likely to be general. Indeed,
the distributions are expected to be similar if the plants procured by park managers are
chosen without regard to their attractiveness to flower-visiting insects. Therefore, the
results strongly suggest that there is considerable scope for making parks and gardens
considerably more insect-friendly through making more judicious choices of the varieties
to be grown. Although insect-attractive varieties are relatively few, our survey identified a
number of them, some of which are not included on the RHS 'Perfect for Pollinators' list
or marked by a pollinator-friendly label (e.g., *Iberis* 'Masterpiece', *Alstromeria* 'Inticancha
Dark Purple' and *Tradescantia* 'Blue and Gold', Table S2).

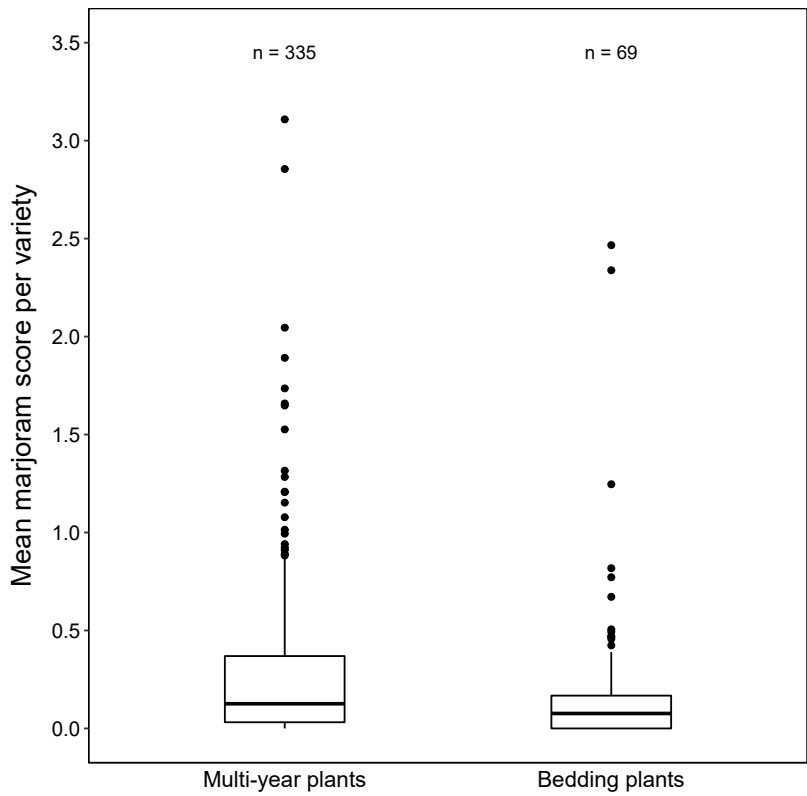

**Figure 5** **Boxplots of mean marjoram scores of varieties in the two plant longevity categories.** Multi-year plants were defined as plants that are typically grown for more than one season in Britain, which includes shrubs and fully-hardy herbaceous perennials. Bedding plants were defined as plants that are typically grown for only one season, which includes annuals, as well as tender and half-hardy biennials and perennials. Data are aggregated from all six study locations.

Although most flower-visiting insects are generalists visiting a wide range of plant species, some are specialists dependent on one or a few related plant species or genera. These plants will not necessarily attract a wide range of insects in large numbers, and as such may not be considered attractive, but they can have a place in gardens, especially if they are aesthetically pleasing and could offer an opportunity to observe and aid insects that are of special interest or are rare (*Garbuzov & Ratnieks, 2014a*).

## Current advice on pollinator-friendliness is imperfect

Our results showed that the median attractiveness of varieties with a bee- or pollinator-friendly symbol on the label, or those present on the RHS 'Perfect for Pollinators' list, was 4.2 times greater than of varieties without such recommendations. However, given the large range and spread of varieties in each category, there was considerable overlap between the three distributions (Fig. 4). In other words, there were many recommended varieties that were unattractive or poorly attractive to insects, and some non-recommended varieties that were very attractive (Table S2). Thus, the use of the word "perfect" by the RHS list is inaccurate and presumably was chosen for marketing reasons. Nonetheless, the recommended plants were about four times more attractive to flower-visiting insects,

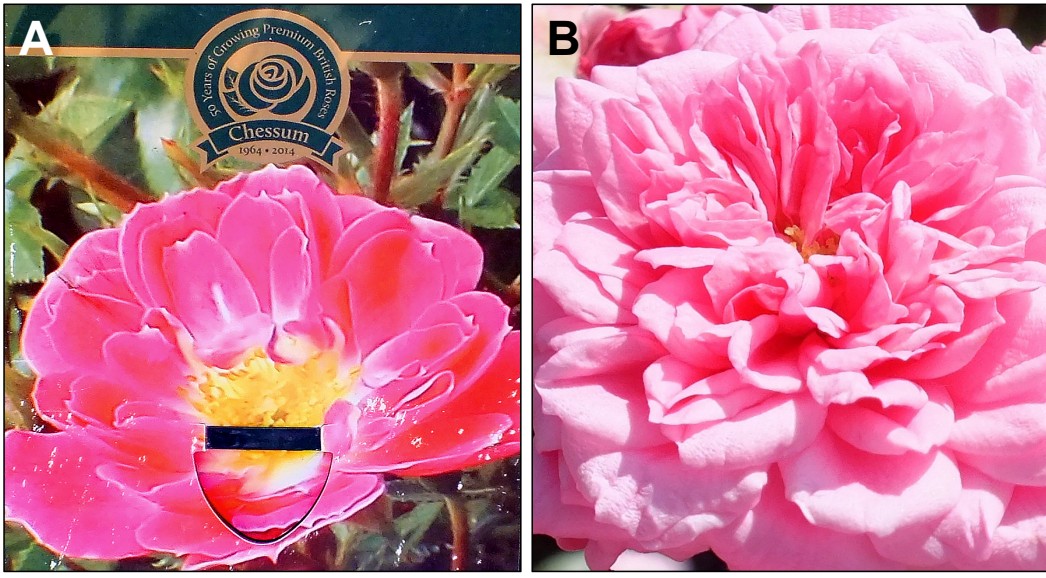

**Figure 6 Discrepancy between plants and their labels.** Patio rose 'Glowing Pink', with the flower displayed on the label (A) considerably different from the actual flowers on the plant (B).

which is a step in the right direction that could further be improved by greater reliance on empirical data, as this becomes available. However, given the two orders of magnitude range in attractiveness typically seen among ornamental plants (this study, *Garbuzov & Ratnieks, 2014b*; *Garbuzov, Samuelson & Ratnieks, 2015*), a three-fold median difference between recommended and non-recommended varieties could potentially be greatly improved. Additionally, the plant labels themselves are not always accurate. For example, the label on the patio rose 'Glowing Pink' showed flowers with a large area of insect-rewarding plant parts in the centre, while the actual flowers on the plant looked markedly different—closed with an excessive number of petals (Fig. 6).

Pollinator-friendly recommendations are generally based on anecdotal experiences and opinions, rather than on empirical evidence (*Garbuzov & Ratnieks, 2014a*), which reduces their value. However, given the tens of thousands of varieties in existence, and the variability of the natural world, it is doubtful that any list of recommendations can ever be definitive. Furthermore, recommendations are often given at a level that is too broad. For example, the RHS 'Perfect for Pollinators' list recommends "*Aster* species and hybrids" (*RHS, 2014*), implying that all species, hybrids and varieties are equally attractive to pollinators. However, even within this one group of plants, the hundreds of different species, hybrids and varieties available to gardeners are hugely variable in their attractiveness (*Garbuzov & Ratnieks, 2015*).

## Multi-year plants are slightly more attractive than bedding plants

Our results show that the median attractiveness of multi-year plants is 1.6 times greater than that of bedding plants. Perennials are generally characterized by higher levels of assimilated resources than annuals, and hence higher amounts are generally allocated to sexual reproduction and pollinator reward (*Corbet, 1995*). There is some evidence that the

abundance of certain bee families, Apidae, Megachilidae and Anthophoridae, is positively associated with plant communities dominated by perennial, as opposed to annual plants, as is characteristic of later successional stages (*Parrish & Bazzaz, 1979*; *Fussell & Corbet, 1992*; *Steffan-Dewenter & Tscharntke, 2001*).

## Garden centres are valuable sources of data

Our results show that flower-visiting insects are attracted in large numbers to plants in full bloom on display for sale to customers in garden centres (Table 2). The most common insect groups seen visiting flowers were bees (predominantly honey and bumble bees) and flies (predominantly hover flies, Syrphidae), in agreement with surveys of insects on plants in gardens in the same region (Table 2, *Garbuzov & Ratnieks, 2014b*; *Garbuzov, Samuelson & Ratnieks, 2015*).

The relative attractiveness of plant varieties was also consistent with that from two previous surveys of plants growing in gardens (*Garbuzov & Ratnieks, 2014b*; *Garbuzov, Samuelson & Ratnieks, 2015*). Percentile ranks of five plant species or groups that were in common between this survey and the previous surveys (Table S1) showed significant correlations close to a 1:1 ratio (Fig. 2A), although these correlations were not expected to be perfect or near 1:1, as the assemblages of plants in each survey were different and insect abundance and diversity would also have differed. Nonetheless, the percentile rank correlations showed that certain plants tended to be among the most attractive (*Origanum*, *Nepeta*, open-flowered *Dahlia* varieties), moderately attractive (*Lavandula*), or least attractive (*Pelargonium*) whether on sale in a garden centre in pots or growing in a garden. Correlations in the absolute mean insect counts per unit area were also strong and significant ($P = 0.001$, *Garbuzov & Ratnieks, 2014b*) or marginally non-significant ($P = 0.089$, *Garbuzov, Samuelson & Ratnieks, 2015*) between this survey and the two previous surveys, although considerably below 1:1 (Fig. 2B), perhaps because plants in pots attract fewer insects than those established in the ground.

Importantly, this shows that plants on sale in garden centres can be surveyed in a meaningful way in relation to plants growing in gardens. In addition, our survey shows that garden centres sell a wide range of ornamental flowering plant varieties, most of which are in full bloom as they are displayed for sale. This is presumably because bloom is the main feature of interest to the customers. However, it is also convenient for conducting surveys of insects visiting flowers.

## CONCLUSIONS

Our study showed that most flowering varieties being sold to the public in England are relatively unattractive to flower-visiting insects. If most private gardeners obtain most of their garden plants from garden centres, then, on the whole, urban gardens are probably not as friendly to flower-visiting insects as they could potentially be. There is considerable scope for making gardens more insect-friendly through judicious choices of varieties that are grown. We have also shown that the plants recommended as friendly to pollinators are only a few times better than the plants that are not recommended, presumably because these recommendations generally rely on casual observations or opinions, and could further be

improved by greater reliance on more rigorous empirical data. Although being a few times better is a step in the right direction for recommended varieties, given the c. 100-fold range in attractiveness among garden plants, there is ample room for improvement. Furthermore, our study demonstrates that garden centres can be valuable sources of data, as they sell a wide range of ornamental flowering plant varieties, most of which are in full bloom when sold. The insect-attractiveness data gathered in garden centres, where the plants are in pots, appear to be consistent with the data from parks and gardens, where the plants are established in the ground.

We suggest there is also a scope for garden centres to be pro-active and take the initiative by carrying out surveys to quantify the insect-attractiveness of plant varieties that they sell, as the methods for doing so are relatively straightforward to follow, in order to make their pollinator-friendly promotions more accurate and evidence-based. Furthermore, customers themselves could make a few casual observations before a purchase, which, although not very accurate due to stochasticity of insect counts, would give a rough indication of plant attractiveness. Thus, customers could exert consumer pressure by creating more demand for varieties attractive to flower-visiting insects, which garden centres could potentially respond to. Garden centres could also use this information to increase the proportion of insect-friendly varieties that are offered for sale, and thus help pollinators in gardens and in the wider environment.

## ACKNOWLEDGEMENTS

We are grateful to Zoe Bruce (Garden Pride), John Leonard (Wyevale), Phil Cottingham (Staverton Nursery), Graham Gough (Marchants Hardy Plants), Michael Edmondson (Hillier), and John Paisley (Wych Cross) for their assistance and for allowing us to carry out our surveys in the garden centres under their management.

### Funding
The authors received no funding for this work.

### Competing Interests
The authors declare there are no competing interests.

### Author Contributions
- Mihail Garbuzov conceived and designed the experiments, performed the experiments, analyzed the data, wrote the paper, prepared figures and/or tables, reviewed drafts of the paper.
- Karin Alton conceived and designed the experiments, performed the experiments, reviewed drafts of the paper.
- Francis L.W. Ratnieks conceived and designed the experiments, wrote the paper, prepared figures and/or tables, reviewed drafts of the paper.

## Data Availability

The raw data has been supplied as a Supplementary File.

## Supplemental Information

Supplemental information for this article can be found online at http://dx.doi.org/10.7717/peerj.3066#supplemental-information.

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
