# Peer review of "Most ornamental plants on sale in garden centres are unattractive to flower-visiting insects"

_PeerJ, doi:10.7717/peerj.3066_

## Round 0.1 · original submission · Minor Revisions

This is an interesting study looking at the availability of pollinator friendly plants in garden centres and whether those plants that are labelled as such are actually pollinator friendly. The reviewers have raised a number of points and, although they are many, I do not believe they will be difficult to fix. I do agree however that the use of supply chain in the title is misleading as you are essentially looking at a single link in the overall chain and would encourage the authors to correct the title to reflect this point.

Reviewer 1 ·

Basic reporting

This study covers a subject that has received little attention and would be a valuable contribution to the literature. With some changes to improve the quality of the manuscript, I feel this study will be adequate for publication in PeerJ.

Title:
I don’t feel that the title accurately represents the content of the article. The study itself does not assess the supply chain of the plants at all, and I suggest this section of the title is removed. The study is also not restricted to wildlife gardening and is concerned with everyday gardening. The title needs to be changed to more accurately reflect the nature of the study.

Introduction:
Line 52: The use of the first citation (Ghazoul, 2015) is wrong – this paper does not say that pollinating insects appear to be in a global decline and actually questions if there is currently enough evidence to justify this assertion. If this citation is to be used, this should be reworded to represent the content of that article, or a more appropriate citation should be used.

Line 62 and throughout: Please define what is meant by ‘attractiveness to flower-visiting insects’ so the reader doesn’t have to refer to a previous publication for a definition.

Line 78: This is a fast moving literature and I would suggest citing some more up-to-date reviews on the topics of the benefits of urban nature, rather than 10 year old studies. E.g.:

Clark NE, Lovell R, Wheeler BW, Higgins SL, Depledge MH, Norris K. 2014. Biodiversity, cultural pathways, and human health: A framework. Trends in Ecology and Evolution 29: 198–204.
Shanahan DF, Fuller RA, Bush R, Lin BB, Gaston KJ. 2015. The health benefits of urban nature: How much do we need? BioScience 65: 476–485.

Line 82: Should say “IS estimated to have attracted 46 million visitors…”

Lines 83-92: I don’t think this paragraph is necessary to outline the importance of the study and is irrelevant to the data collected. Why do people need to know about the other things that garden centres sell?

Experimental design

Methodology:
Lines 110-112: This information, about the surveys being undertaken on two consecutive days, is repeated more fully on lines 128-129 so is not needed here as well.
Line 122-127: The distance of the survey sites to the nearest patch of natural vegetation is potentially important for floral resources – was this measured? Could this information be somehow incorporated into the analysis? If not, this potential flaw should be expanded upon in the discussion.
Line 126: Missing ‘(ref)’ for an important point about flower patch size not affecting the number of insects attracted per unit area. Without this reference, it is impossible to verify this point.

Please report who carried out the surveys and their level of expertise, and if the same individuals carried out the surveys at all sites.

Table 1: Dates surveyed – this study took place over the season, could this be a reason for the variation you find between the insect counts in the different garden centres? Was this explored? If so, this should be expanded upon in the manuscript, or if not, this should be addressed in the discussion.

Validity of the findings

Results:
Line 189: A result cannot be ‘marginally non-significant’. It is either significant or not significant at the threshold set when testing the hypothesis (0.05).
Line 207 and throughout: Be consistent with the number of decimal places reported.

Discussion:
This section should be restructured so that the main findings are discussed first, i.e. that most plants were relatively unattractive to flower-visiting insects, and that current advice on pollinator friendliness is imperfect. I would suggest moving the section about the use of garden centres as a source of data towards the end of the section.
An aspect not discussed in this manuscript but I think warrants attention is to look at the level to which the plant and insect species are specialists and generalists – although some plants may be very attractive to a high number of individual insects, some may be attractive to just one or two specialist species and therefore may have received fewer visitors, however, these plant species would be just as important to encourage to be planted to create a diverse urban ecosystem.
I understand this is beyond this particular dataset, but it would be of further interest to understand if there is a relationship between human aesthetic appreciation and attractiveness to pollinators (for example see the work of Lindemann-Matthies) and I feel this would be worth mentioning in this discussion:
Lindemann-Matthies P, Marty T. 2013. Does ecological gardening increase species richness and aesthetic quality of a garden? Biological Conservation 159: 37–44.
Lindemann-Matthies P, Junge X, Matthies D. 2010. The influence of plant diversity on people’s perception and aesthetic appreciation of grassland vegetation. Biological Conservation 143: 195–202.

Finally, if the data is available, it would be interesting to see if there is any relationship between the cost of the plants and the attractiveness to flower-visitors, as I expect this would be a better predictor of people’s choice of garden plant species. It would also be important in future studies to understand the factors which make up gardeners choice in species, and how much this is impacted by their understanding of their value to pollinators, the monetary cost, and how attractive they found the species. Do the pollinator-friendly accreditations actually make a difference?

Conclusions:
Much of this repeats points made in the discussion and should be restructured in the same way as suggested above.
Line 316: What does ‘are only a few times better than the plants that are not recommended’ mean? Surely being multiple times better is a good thing. This needs to be explained.

Acknowledgements:
Names have been left out, with “[Names]” in brackets. This shows a lack of attention to detail.

Figures and tables:
Figure 1: This is fine to include as supplementary material, but I don’t see what this brings to the manuscript or why the reader needs to see what pollinator-friendly logos look like and would recommend removing from the main paper.
Figure 2: This is a repeat of data that is found already in table 2. This doesn’t need to be displayed twice. Also, Pie Charts are not an adequate way of displaying this information and could be difficult to interpret and compare between, if this must be presented than a series of bar charts would be more appropriate. Also please check that the colour pallet is possible to differentiate for colour blind people, as many readers may be colour blind.
Figure 4: Is it necessary for the reader to see these graphs? They contain a lot of repeated information (for example the number of varieties sampled which can be found in table 1). They may be more suited to put as supplemental material. Also, although mentioned in the figure description, the meaning of the black labelling is not included in the legend with the colours and this requires fixing if this figure is to be kept.
Figure 5: I don’t understand the description ‘lowercase letters above boxplots indicate significant pairwise differences…’ I do not see any such letters and this is confusing to the reader.
Figure 7: The same as my comment to figure 1.

Reviewer 2 ·

Basic reporting

No Comments

Experimental design

- I liked the experimental design and the fact that it was simple but appropriate and well thought out
- By bringing their own Origanum, the authors dealt with one of the major problems of studying pollinators in the field, across sites, which is establishing a baseline.
- I liked the fact they surveyed 6 garden centres, and not all from the same chain and varying in size.
- Robust stats
- Clear figures and diagrams

Validity of the findings

I think this is a valid study and sheds light on what one often suspects is a slightly random assignment of "pollinator friendly" logos to plants. If companies are to sell plants off the back of the public's desire to attract pollinators then they should be properly informed with empirical data. This is clearly part of an on-going body of work which is designed to investigate ornamental garden plants as resources to pollinators, which is an important contribution to pollinator conservation, seeing as decline in resources is frequently cited as a cause of declining pollinator abundance and diversity.

Additional comments

I hope the authors use this work to inform and guide garden centres so as to inform and enable consumers to make the best choices if their aim is to purchase plants that are beneficial to pollinators.

Reviewer 3 ·

Basic reporting

Policies:
- Ethics Policy: I may be wrong but I think this counts as a 'field study'. However, I did not see an ethics statement in the MS.
- Do the authors have permission to reproduce the logos of the RHS and other labels in Figure 1 of the publication (e.g. the RHS Perfect for Pollinators logo is a registered trademark )?

Writing style
- The writing style is clear and professional on the whole but there are several places where the meaning is unclear, where the writing is overly complicated or not obviously relevant, or where it seems that the text has not been checked for errors.
o The introduction starts strongly with some well-referenced text on knowledge gaps and important, then between L79-92, there is a lot of information about the non-plant products available at, and the social importance of, garden centres. I don’t find this much detail relevant to the overall research questions and I think this could be shortened considerably or removed. Is there a reference that shows that real-world garden centres are still visited a lot in the age of online sales? That may make the point better.
o L121: I had to read this a couple of times to understand. Reword this to ‘This was between 59 and 74 varieties covering between 14.7 and 37.6 m2 per garden centre”
o L177: was this the same one of six garden centres or different ones each time?
o L278 onwards: The wording of this section is not as professional as it could be,. The RHS states itself that the logo is suitable for “plants that support pollinating insects”.
o L339 – names have been omitted from the acknowledgements.

Referencing
There were several places where referencing could have been improved:
- L63 states "For example, asters are often recommended as pollinator-friendly." but there is not a reference for this.
- - L126 just states "(ref)"
- L158-164 references needed, if not for all of the plant categories then just for the hardiness rating
- L234-239: the previous surveys should be referenced here as well as the supporting information table.
- Table 1: This seems like a lot of information could some of it (e.g. the addresses) go in supporting information?

Figures
Figure 1: Not high resolution
Figure 2: This is difficult to understand easily and compare between different charts. A stacked bar chart may be better or colours should be changed to make it clearer (and also to avoid problems for those with red/green colour blindness).

Experimental design

Clearly defined research question
-By the end of the paper I understood the relevance of the research but for most of the paper I was not sure why this work was being carried out, or how it may have application/real world utility. I think it would help to go through the introduction and conclusion and clearly define the need for this work. In particular:
- L93-103 could have been a good opportunity to justify the need for this research/place it in context of previous studies, but no literature is cited.
- A summary of the key findings and why they are important at the start of the Discussion would be useful.
- It was quite hard to understand the links between the two questions of whether garden centres have plants attractive to insects, and whether plants labeled as good for pollinators actually are. Make the links clearer if possible.

Ethics
As mentioned above there is no ethical statement but as the authors clearly had permission from the garden centres, it is likely that ethical standards were followed.

Methods
- The survey methodology is not completely reproducible
o L129: 12 counts for insects were carried out but it is not stated at what intervals
o L131: how were they identified? Were photographs used or was this by eye? How many people were identifying the insects to ensure nothing was missed/errors were avoided?

Validity of the findings

Conclusions
- The conclusions include statements that relate the findings to parks and gardens, but I don’t see any evidence provided that professional gardeners buy plants from commercial garden centres such as these.
- The lack of referencing in some parts (As noted above) made some of the statements appear to be speculative, but this may not be the case.
- L263-265: this doesn’t seem to be a conclusion from the results in this paper.
- L320: Is there anything to support this assumption about consumer behaviour? Is there any literature to suggest that more choice of insect friendly plants would mean more people would choose to buy them?
- L328 onwards: Is it likely that consumers would do this? How could people create demand if the products are not there to buy?

---

## Round 0.2 · accepted · Accept

Thank you for submitting the corrections and the detailed rebuttal letter. I'm happy that I can now accept the paper.